# Never Skip a Batch: Continuous Training of Temporal GNNs via Adaptive Pseudo-Supervision

## Abstract

Temporal Graph Networks (TGNs), while being accurate, face significant training inefficiencies due to irregular supervision signals in dynamic graphs, which induce sparse gradient updates. We first theoretically establish that aggregating historical node interactions into pseudo-labels reduces gradient variance, accelerating convergence. Building on this analysis, we propose History-Averaged Labels (HAL), a method that dynamically enriches training batches with pseudo-targets derived from historical label distributions. HAL ensures continuous parameter updates without architectural modifications by converting idle computation into productive learning steps. Experiments on the Temporal Graph Benchmark (TGB) validate our findings and an assumption about slow change of user preferences: HAL accelerates TGNv2 training by up to $13\times$ while maintaining competitive performance. Thus, this work offers an efficient, lightweight, architecture-agnostic, and theoretically motivated solution to label sparsity in temporal graph learning.

## 1 Introduction

Temporal graphs represent evolving interactions between entities over time. Their models capture complex patterns, improving predictions in diverse applications spanning from recommendation systems to financial fraud detection Deng et al. (2019); Song et al. (2019); Zhao et al. (2019).

Temporal Graph Networks (TGNs) solve the problem with the state-of-the-art performance Rossi et al. (2020); Tjandra et al. (2024). Even for TGNs, the training is impeded by the sparsity of supervision signals. Node-level labels (e.g., explicit user preferences or interactions) appear irregularly, leaving many time steps unlabeled. Current TGN implementations process batches in two modes: full training steps for batches with supervision labels and memory-related state updates for others. This creates an efficiency bottleneck — as labeled batches constitute less than $2\%$ of interactions for many applied problems Gastinger et al. (2024). As a result, models skip parameter updates for large portions of the data, slowing convergence.

For most temporal node prediction datasets, the target is forecasting interaction type, such as a music genre, a subreddit, or a token Huang et al. (2023) within a specified time frame. Such interactions are a realization of a marked temporal point process, such as a non-homogeneous Poisson process, with a dynamics governed by a slowly changing latent state of an object Cai et al. (2018). Although the latent state may stay stable for long intervals, in practice it typically changes gradually Klenitskiy et al. (2024); Li et al. (2024). Such slow evolution provides temporal consistency that models can leverage: from temporal-neighbor aggregation Trivedi et al. (2019) to Temporal Graph Networks with learned memory Rossi et al. (2020) that achieved state-of-the-art results through learned memory modules. Further discussion of the related work is available in Appendix A.

Given the slow dynamics of node preferences, we propose a history-based pseudo-labeling scheme based on the moving average idea (MA) that turns unlabeled batches into supervised training signals, achieving faster training while maintaining model quality for TGNs. The MA operates orthogonally to model architecture choices and introduces no additional parameters. By augmenting the standard cross-entropy loss with pseudo-labeled batches, we transform previously idle computation into productive training steps. Our work shifts the paradigm from "train only when supervised" to "always train, intelligently extrapolate," offering a principled solution to gradient sparsity in dynamic graph

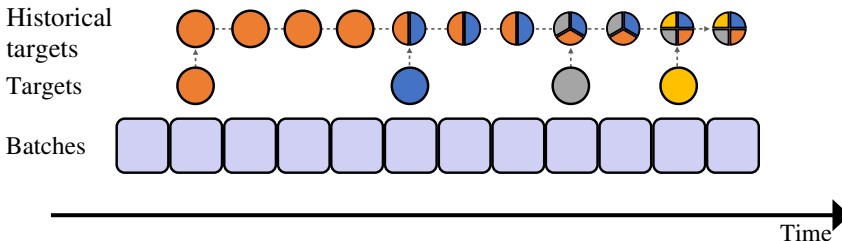

Figure 1: Our training with historical pseudo-labels. We extend beyond the vanilla method's sparse real ground truth supervision (Targets, the bottom row of circles) by generating pseudo-labels derived from historical label patterns (Historical targets, HA, the top row of circles), enabling more comprehensive training data utilization.

learning. The training pseudo-labels for MA is the exponentially weighted recent targets. We also compare MA to two alternative pseudo-labels generation strategies, Persistent Forecast, that uses *last-observed labels*, and Historical average, that uses *window-based averages of past labels*.

Comprehensive experiments on the Temporal Graph Benchmark (TGB) Gastinger et al. (2024) validate our approach. Our experiments show consistent acceleration of model convergence by 2-13x across all four benchmark datasets, while maintaining competitive prediction accuracy and even improving the final result in some cases.

To summarize, the main claims of this paper are:

- **History-based pseudo-labeling approach.** We propose a pseudo-labeling based on Exponential History Moving Averaging (MA) suitable for temporal graphs, reducing label sparsity.

- **Proof of better convergence given HAL.** Under the constant user preferences assumption, we prove faster convergence for history-based aggregation of labels with the theoretical increase of speed $\min(h, k)$, where $h$ is the aggregated history length and $k$ is the number of possible interactions associated with a user node.

- **TGN architectures equipped with MA.** We implemented MA for training both TGNv2 Tjandra et al. (2024) and a modified DyRep Trivedi et al. (2019) model (DyRep v2) and showed that it reduces training time up to 13 times without degradation of model quality working better than other aggregation strategies. A study confirms that the improvement is connected with the intrinsic temporal dynamics for 2 out of 4 considered datasets, where the appearance order for events can be ignored — further motivating HAL aggregations and inclusion of additional temporal benchmark with more evident temporal dynamics of preferences.

## 2 METHOD

### 2.1 GENERAL PIPELINE

Let $G = (V, E)$ be a temporal graph with vertices $V$ and temporal interaction edges $E$. Each timestamped edge $e = (u, v, t, \mathbf{f}_e)$, where $u, v \in V$ are source and destination nodes, $t \in \mathbb{R}^+$ is a timestamp, and $\mathbf{f}_e \in \mathbb{R}^d$ is a vector of edge features.

Thus, edges appear and vanish with time, making their prediction a vital problem. Formally, for a subset of nodes $V' \subseteq V$, $|V'| = n$ (e.g., users), we predict time-varying affinity toward other nodes or categories (e.g., items, music genres). The target $\mathbf{y}_t^{(v)} = (y_{t,1}^{(v)}, \ldots, y_{t,n}^{(v)}) \in \Delta^n$ for node $v \in V'$ is then an $n$-dimensional probability vector representing normalized preferences at time $t$ where $\Delta^n = \{\mathbf{y} : \sum y_i = 1, y_i \geq 0\}$ (as defined in Section 3.1). Each element $y_{t,i}^{(v)} \geq 0$ quantifies the affinity of $v$ toward the $i$-th category at time $t$. These targets are observed irregularly, with most vectors lacking ground-truth affinities.

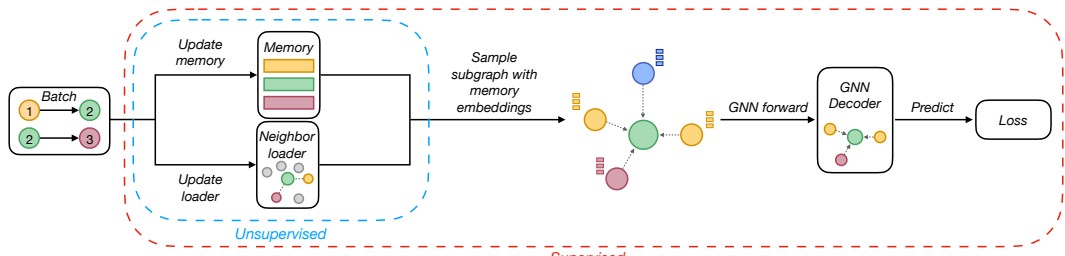

Figure 2: Comparison of batch processing pipelines: The unsupervised pipeline (blue dashed box) performs memory and neighbor loader updates only, while the supervised pipeline (red dashed box) encompasses these steps and extends them with subgraph sampling, GNN processing, and loss computation for model training.

During training, the edges are processed chronologically and divided into sequential batches $\{B_1, \ldots, B_T\}$, where each batch $B_t$ contains a fixed number $N$ of consecutive edges $\{e_{t_1}, \ldots, e_{t_N}\}$ that correspond to time moment $t$. As targets for a single temporal edge are rarely available, batches also contain sparse labels. This sparse supervision creates the fundamental challenge we address: labeled batches may represent less than 2% of interactions, leading to the gradient sparsity problem analyzed in Section 3.

In the Temporal Graph Network framework, batch processing operates in two distinct modes depending on the presence of supervision targets within the batch.

**Unsupervised batch processing.** When a batch $B_t$ arrives without supervision targets, the model performs only memory updates, without any gradient-based training. Temporal edges in $B_t$ are used to update node memories by aggregating interaction history. The neighbor sampler is also refreshed to incorporate these new interactions for future subgraph construction. Since no ground-truth labels are available, no loss is computed, and model parameters remain unchanged during this step.

**Supervised batch processing.** When $B_t$ contains supervision targets $\{\mathbf{y}_t^{(v)}\}$, the model performs a full training step, which includes both memory updates and gradient-based optimization.

First, temporal edges in $B_t$ that occurred before the target time are used to update node memories. This ensures that predictions are based on historical information only (i.e., no information leakage from the future). The neighbor sampler is also updated accordingly.

Then, for each target node $v \in V'$, the following steps are performed:

1. Temporal subgraph sampling: A subgraph is extracted using the updated sampler, constrained to edges occurring before time $t$.

2. Initial embeddings: The memory module provides time-aware node embeddings reflecting their historical state.

3. Forward pass: A graph neural network (GNN) processes the subgraph to compute context-aware node embeddings.

4. Prediction and loss: The model generates task-specific predictions, which are compared to the ground-truth labels $\mathbf{y}_t^{(v)}$. The resulting loss $\mathcal{L}$ is computed.

5. Backpropagation: The loss is backpropagated to update: GNN parameters (e.g., message-passing layers, attention mechanisms) and memory-related parameters (e.g., GRU or RNN weights controlling memory updates over time).

The complete workflow for processing batches is illustrated in Figure 2, showing how the pipeline integrates memory updates with neural network training.

The model $f_\theta$ processes edge sequences to update node embeddings $\mathbf{h}_v^{(t)}$ and predict target labels $\hat{\mathbf{y}}_t^{(v)}$, with loss computed over all active nodes.

$$\mathcal{L}(\boldsymbol{\theta}) = \frac{1}{T} \sum_{t=1}^{T} \sum_{v \in \mathcal{V}_t} \mathcal{L}_{\mathrm{CE}} \left( \hat{\mathbf{y}}_t^{(v)}, \mathbf{y}_t^{(v)} \right) \to \min_{\boldsymbol{\theta}}, \tag{1}$$

where $\mathcal{L}_{\mathrm{CE}}$ denotes the cross-entropy loss. In our loss function $\tilde{\mathcal{L}}$, we replace $\mathbf{y}_t^{(v)}$ with pseudo-labels $\tilde{\mathbf{y}}_t^{(v)}$, that are non-zero for almost all nodes and batches.

## 2.2 GENERATION OF HISTORICAL PSEUDO-LABELS

For each batch $B_t$ we compute pseudo-targets $\tilde{y}_t^{(v)}$ *only for nodes $v$ participating in $B_t$*. The core component $\tilde{y}_t^{(v)}$ is an aggregate of all historical supervision signals observed for node $v$ prior to $t$,

$$\tilde{\mathbf{y}}_t^{(v)} = \bar{\mathbf{y}}_t^{(v)} + \gamma \cdot \epsilon, \tag{2}$$

where $\bar{\mathbf{y}}_t$ aggregates past targets, and $\epsilon$ is noise generated by first sampling $\epsilon' \sim \mathcal{U}(-\alpha, \alpha)$ and then subtracting the mean to ensure $\sum_i \epsilon_i = 0$. This guarantees that the sum $\sum_i \tilde{y}_{t,i}^{(v)} = 1$. Here, $\gamma \geq 0$ is a noise scaling factor controlling the magnitude of regularization. This formulation ensures that $\tilde{y}_{t,i}^{(v)}$ remains a valid probability distribution while reducing gradient variance as proven in Theorem 3. We can define $\bar{\mathbf{y}}_t$, derived from $v$'s past observations, in different ways. This paper considers the following aggregation strategies.

**Historical Average (HA).** Aggregates all targets for $v$ across previous batches:

$$\bar{\mathbf{y}}_t^{(v)} = \frac{1}{L^{(v)}} \sum_{t' < t} \mathbf{y}_{t'}^{(v)}, \tag{3}$$

where $L^{(v)}$ is the number of observed targets for a node $v$. This setting corresponds to the HAL case in Subsection 3.2, where the user affinities remain constant. Below, we provide two options that make weaker assumptions about the correlation between the current user preferences and the past observed labels: Moving average and Persistent forecast.

**Moving Average (MA).** Updates the pseudo-label incrementally when new targets arrive. Suppose a new target $\mathbf{y}_t^{(v)}$ is observed for $v$ at batch $t$. The updated pseudo-label becomes:

$$\bar{\mathbf{y}}_t^{(v)} = \frac{w-1}{w} \bar{\mathbf{y}}_{t-1}^{(v)} + \frac{1}{w} \mathbf{y}_t^{(v)}, \tag{4}$$

with $w$ being a method hyperparameter. Initially, $\bar{\mathbf{y}}_0^{(v)} = \mathbf{y}_0^{(v)}$.

**Persistent Forecast (PF).** Reuses the most recent observed target for $v$:

$$\bar{\mathbf{y}}_t^{(v)} = \mathbf{y}_\tau^{(v)}, \quad \tau = \max\{t' \leq t \,|\, \mathbf{y}_{t'}^{(v)} \text{ is available}\}. \tag{5}$$

Updating introduced aggregations over timestamps is efficient and can be done iteratively over batches. Historical Average and Moving Average take into account all past labels, while weighting them differently, while Persistent Forecast uses the last available value. Also, both $\bar{\mathbf{y}}_t^{(v)}$ and $\tilde{\mathbf{y}}_t^{(v)}$ are valid probability distribution. For $\bar{\mathbf{y}}_t^{(v)}$ for MA, it easily follows from the induction rule, for HA it also holds, as it uses a mean value of correct distributions, and for PF it is evident. For $\tilde{\mathbf{y}}_t^{(v)}$, the introduced normalization procedure leads to the desired effect.

# 3 STOCHASTIC GRADIENT DESCENT CONVERGENCE FOR HISTORICAL LABEL AVERAGING

In this section, we present our main theoretical result on the convergence speed for the pseudo-label generation approach based on Historically Aggregated Labels (HAL). We start with the introduction of preliminaries on the convergence of Stochastic Gradient Descent (SGD). Then, the theorem describes an upper bound for the convergence rate of HAL under natural assumptions and compare it with a vanilla variant. Complete proofs of the presented results are available in the Appendix B.

## 3.1 PRELIMINARIES

Let $D = \{(\mathbf{x}_i, \mathbf{y}_i)\}_{i=1}^m$ be i.i.d. samples drawn from an unknown distribution $\mathcal{D}$. $\mathbf{x} \in \mathcal{X} \subseteq \mathbb{R}^{d_x}$, $\mathbf{y} \in \mathcal{Y} \subseteq \mathbb{R}^n$. Below, $\mathcal{Y} = \Delta^n = \{\mathbf{y} : \sum y_i = 1, y_i \geq 0\}$ is an $n$-dimensional simplex for a classification problem. For one-hot encoding a single component, associated with the true label, $y_i = 1$ and all others are zero.

Given the sample $D$, the empirical risk minimization problem for the parameter vector $\boldsymbol{\theta} \in \mathbb{R}^d$ is:

$$\mathcal{L}(\boldsymbol{\theta}) = \frac{1}{m} \sum_{i=1}^m \ell\left(\boldsymbol{\theta}; (\mathbf{x}_i, \mathbf{y}_i)\right) \to \min_{\boldsymbol{\theta} \in \mathbb{R}^d}, \tag{6}$$

where $\ell : \mathbb{R}^d \times \mathcal{X} \times \mathcal{Y} \to \mathbb{R}_+$ is a differentiable loss function.

Stochastic Gradient Descent (SGD) Robbins & Monro (1951) updates the parameter vector, starting from the initialization $\boldsymbol{\theta}_0$, via

$$\boldsymbol{\theta}_{t+1} = \boldsymbol{\theta}_t - \alpha_t \mathbf{g}_t, \qquad \mathbf{g}_t = \frac{1}{B} \sum_{j \in \mathcal{B}_t} \nabla \ell\left(\boldsymbol{\theta}_t; (\mathbf{x}_j, \mathbf{y}_j)\right), \tag{7}$$

where $t = 0, \ldots, T-1$ is the SGD iteration number, $\mathcal{B}_t$ is a batch of size $|\mathcal{B}_t| = B$ sampled without replacement. The estimator $g_t$ is unbiased, $\mathbb{E}[g_t | \boldsymbol{\theta}_t] = \nabla \mathcal{L}(\boldsymbol{\theta}_t)$, but exhibits non–zero variance

$$\sigma^2 = \mathbb{E}\|g_t - \nabla \mathcal{L}(\boldsymbol{\theta}_t)\|^2 = \frac{1}{B} \mathbb{E}_{(\mathbf{x}, \mathbf{y}) \sim \mathcal{D}} \|\nabla \ell(\boldsymbol{\theta}_t; (\mathbf{x}, \mathbf{y})) - \nabla \mathcal{L}(\boldsymbol{\theta}_t)\|^2. \tag{8}$$

Within this framework, we substitute the original loss function. Specifically, in $\ell(\boldsymbol{\theta}_t; (\mathbf{x}, \mathbf{y}))$ our method would replace $\mathbf{y}$ with a pseudo-labels vector $\mathbf{y}'$. As long as it is unbiased and we can estimate the variance of it, the theoretical results on convergence speed below hold, as one can easily see from Rakhlin et al. (2012); Shalev-Shwartz & Ben-David (2014).

For vanilla SGD, there exists an upper bound a regret $R_T$ for a diminishing step size $\alpha_t = \frac{1}{\mu t}$:

$$R_T = \mathbb{E}\left[\mathcal{L}(\boldsymbol{\theta}_T) - \mathcal{L}(\boldsymbol{\theta}^\star)\right],$$

where $\boldsymbol{\theta}^\star$ denotes the unique minimizer of $\mathcal{L}$.

**Theorem 1** (adopted from Shamir & Zhang (2013)). *For a $\mu$-strongly convex loss function and an unbiased $\mathbf{g}_t$ defined in equation 7 with variance $\sigma^2$ and batch size $B$ for the step size $\alpha_t = \frac{1}{\lambda t}$, the regret has the upper bound:*

$$R_T \leq \frac{17\sigma^2}{\mu B T} \left(1 + \log T\right). \tag{9}$$

This bound is sufficiently tight to describe the real dynamics Shalev-Shwartz & Ben-David (2014). Variations of this bound and SGD are also discussed in literature Bottou et al. (2018); Bubeck (2015), while the above form would be sufficient for our purposes.

The upper bound $\sigma^2(1 + \log T)/(\mu B T)$ depends linearly on the gradient–noise variance $\sigma^2$ and the inverse of the batch size. Consequently, we can adjust the convergence speed by minimizing noise variance $\sigma^2$ and maximizing the batch size. Increasing the batch size $B$ is a common advice for faster convergence, e.g., You et al. (2018) shows that one can train a ResNet model within 20 minutes using ImageNet with large batches. Momentum-based approaches for SGD also indirectly increase batch size, improving convergence Kingma & Ba (2015). However, the second component of the variance reduction, $\sigma^2$, related to searching for a lower-variance noise, is often overlooked.

## 3.2 CONVERGENCE SPEED FOR HISTORY-AVERAGE-LABEL SGD

In this subsection, we present our results on the convergence of SGD with historical pseudo-labels and demonstrate its improvement over a vanilla variant of SGD. Formal proofs for the statements are presented in the Appendix.

Suppose that there are $k$ out of $n$ true labels with a uniform probability of occurrence over them. Each time, a user selects a label uniformly at random to produce a single label. In this case, we can

consider an alternative label for a single observation that aggregates past labels to produce $\mathbf{y}$ at the current moment — and that we would define as history-average labels. In this section, we calculate the variance of the gradients, $\sigma^2$, by decomposing it into the product of the parameter values and the variance related to labelling, which we can obtain in a closed form.

Canonically, for multilabel classification $\mathbf{y}$ is a one-hot vector with $1$ being at the place of the observed label and $0$ at all other places. Let us call it *the one-hot label (OH)*. Aggregating over history and normalising leads to the ground truth vector of the form $\mathbf{y} = (\frac{k_1}{h}, \ldots, \frac{k_n}{h})$, where $h$ is the length of the history of observations and $k_i$ is the number of observations of the $i$-th label within it. We call it *History Average Labels (HA)*. Below we consider two options to present $y_i$ — the $i$-th component of $\mathbf{y}$ vector that belongs to the set of the true labels.

**OH case.** In this scenario, our random variable $y_i = t_1$:

$$t_1 = \eta\xi,$$

where $\eta$ is the event of observing a specific true label, a Bernoulli random variable $\mathrm{Be}(\frac{1}{k})$ and $\xi$ is another Bernoulli random variable, that corresponds to the observation of any of the true labels, it is $\sim \mathrm{Be}(u)$, with $u$ is typically close to $1$.

**HA case (ours).** Now let us consider the aggregation of history. We assume equal probabilities for each of $k$ correct labels $\frac{1}{k}$ and the history of size $h$. Then the presented $y_i = t_h$:

$$t_h = \eta_k\xi,$$

where $\xi$ is defined above and $\eta_k$ is a component of a multinomial random vector with equal probabilities $\frac{1}{k}$ and the total number of observations $h$, divided by $h$, as we aim to match the event type probability. Formally, $t_1$ can be obtained as $t_h$ for $h = 1$. Below, we assume that we form batches from independent observations of HA by considering users separately. In this case, the overall bound for SGD would have the form from equation 9.

**Lemma 2.** *The expectation of $t_h$ is $\frac{u}{k}$ and the variance of $t_h$ is $u\frac{k-1}{k^2h} + u(1-u)\frac{1}{k^2}$.*

For the OH and the HA, we have the same unbiased mean value $\frac{u}{k}$, but the variances differ: the OH's variance is $\sigma^2 = \frac{u}{k}\left(1 - \frac{u}{k}\right)$, and the HAL's variance is $\sim \frac{u}{kh} + \frac{u(1-u)}{k^2}$. Without compromising tightness, one can upper bound the variance of $t_1$ by $\frac{u}{k}$ and of $t_h$ by $\frac{u}{kh} + \frac{u}{k^2}$. For a large history length aggregated $h$ or a large number of items in a catalogue $k$, we have lower variance for labels $\sim \frac{1}{k\min(h,k)}$ compared to the order $\frac{1}{k}$ for the first case. Thus, as $y_i = t_h$ we know the variance for it.

Finally, we need to derive *the variance of the gradient* with respect to the parameters, given the variance in labels, derived above. Let us consider the last layer before the softmax function. It has the form $\mathbf{p} = \mathrm{softmax}(C\mathbf{e})$, where $\mathbf{e}$ is the embedding vector for the last layer, $C$ is the parameters matrix, and $\mathbf{p}$ is the vector of predicted probabilities for labels. For an index that corresponds to the correct label $y$ for the cross-entropy loss function, the gradient is $\frac{\partial \mathcal{L}}{\partial c_{ij}} = (p_i - y_i)e_j$. Thus, the variance of the partial derivative $\mathrm{var}\left(\frac{\partial \mathcal{L}}{\partial c_{ij}}\right) = e_j^2\mathrm{var}(p_i - y_i) = e_j^2\mathrm{var}(y_i) = e_j^2\mathrm{var}(t_h)$. The main term here is the variance $\mathrm{var}(t_h)$. For the previous layers, due to the chain rule, we have a similar linear decomposition of the variance of the gradient of the form $\tilde{c}\mathrm{var}(t_h)$, where $\tilde{c}$ is a non-stochastic constant. Thus, the overall variance of the gradient can be represented in the form $\tilde{c}\mathrm{var}(t_h)$ for some positive constant $\tilde{c} > 0$.

Plugging in our variance estimates from Lemma 2 for $h = 1$ and $h > 1$ into the gradient variance with respect to the parameters, we obtain the convergence speed for OH and HA cases.

**Theorem 3.** *Consider SGD in settings from Theorem 1. The following inequalities for the regret $R_T$ hold for a positive constant $c$:*

- *Under the assumptions of OH, for the regret $R_T$ it holds:*

$$R_T \leq \left(1 - \frac{u}{k}\right)\frac{u}{k}\frac{c}{\mu B}\frac{1 + \log T}{T} \leq \frac{u}{k}\frac{c}{\mu B}\frac{1 + \log T}{T}.$$

- *Under the assumptions of HA, for the regret $R_T$ it holds:*

$$R_T \leq \left(\frac{k-1}{kh} + \frac{1-u}{k}\right)\frac{u}{k}\frac{c}{\mu B}\frac{1 + \log T}{T} \leq \frac{2}{\min(h,k)}\frac{u}{k}\frac{c}{\mu B}\frac{1 + \log T}{T}.$$

With the rightmost more informal upper bounds for each $R_t$ obtained by upper bounding the colored terms in the left inequalities, we see that the convergence speed for HA increases by a factor of $\min(h, k)$ when the history is used. From the theorem above, we can also conclude that for small $k$, an increase of $h$ doesn't affect the convergence speed, as in $\min(h, k)$, $k$ would dominate. Given this results, we theoretically justify a label-aggregation approach for better convergence of TGN models.

## 4 RESULTS

### 4.1 EXPERIMENTS SETUPS

**Datasets and protocol.** TGB Huang et al. (2023) is a recent benchmark for Dynamic Node Property Prediction. We employ all four large-scale dynamic graphs from this benchmark, with varying structural patterns and the number of interactions up to $2.5$ million. The datasets span diverse domains: *tgbn-trade* (international agriculture trade, 1986-2016), *tgbn-genre* (user-music interactions), *tgbn-reddit* (user-subreddit activity), and *tgbn-token* (cryptocurrency transactions). As the target metric we use NDCG@10 with higher values indicating better model quality. Comprehensive dataset statistics, including node counts, edge distributions, and label density metrics as well as expanded discussion on NDCG@10 properties, are given in Appendix C. The most sparse dataset is *tgbn-token* with label sparsity $0.06\%$.

Following the TGB Huang et al. (2023) protocol, datasets are split chronologically into training (70%), validation (15%), and test (15%) sets. For *tgbn-genre*, *tgbn-reddit*, and *tgbn-token*, we further refine the training regime by retaining only the last 5% of chronologically ordered edges in the original training set. Validation and test sets remain unchanged.

This 5% adjustment addresses a critical challenge: the full training sets for these datasets are excessively large, causing models to converge within a single epoch. Such rapid convergence obscures the impact of pseudo-labeling strategies, which aim to accelerate training under label scarcity. By truncating training data to the most recent 5% of interactions, we create a regime where models cannot rely on memorization, thereby emphasizing the role of pseudo-labels in extrapolating temporal trends and mitigating label sparsity. The 5% threshold was empirically determined through ablation studies, balancing the need to simulate sparse supervision while retaining sufficient signal for meaningful learning.

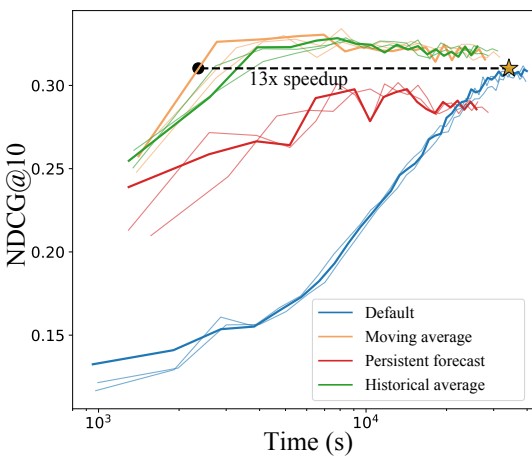

Figure 3: NDCG@10 progression versus logarithmic training time for different pseudo-label strategies on the `tgbn-token` dataset using the TGNv2 model. The x-axis shows training time in seconds (log scale), while the y-axis displays NDCG@10 on valid split. The bolder curves is for the average over three seeds, the lighter ones are for individual learning trajectories.

**Implementation.** TGNv2 Tjandra et al. (2024) is the only architecture achieving meaningful performance on TGB; others collapse to trivial solutions. We focus experiments on TGNv2 and additionally modify DyRep Trivedi et al. (2019) with TGNv2's source-target identification (DyRep v2) to demonstrate architecture-agnosticism. We evaluate three aggregation strategies: Moving Average (MA), Persistent Forecast (PF), and Historical Average (HA), against the Default baseline.

### 4.2 MAIN RESULTS

Table 1 provides a comparison of model performance and efficiency after a single training epoch. To ensure a more consistent baseline in terms of runtime, we additionally include "Default-X" con-

Table 1: Performance after one epoch of training on test split. For greater consistency, we also include Default-X configurations, which show vanilla model training for X epochs. For trade X = 4, for genre, reddit and token X = 2. Results averaged over three seeds with standard deviations available in Appendix E.

| Dataset | Model | NDCG@10 Test ↑ | | | | | Time(s) ↓ | | | | |
|---|---|---|---|---|---|---|---|---|---|---|---|
| | | Default | Default-X | HA | MA | PF | Default | Default-X | HA | MA | PF |
| trade | TGN v2 | 0.388 | 0.469 | 0.669 | **0.729** | 0.710 | 23 | 84 | 74 | 78 | 68 |
| | DyRep v2 | 0.387 | 0.448 | 0.668 | **0.734** | 0.711 | 26 | 102 | 91 | 94 | 81 |
| genre | TGN v2 | 0.363 | 0.392 | 0.419 | **0.421** | 0.389 | 177 | 355 | 233 | 333 | 259 |
| | DyRep v2 | 0.363 | 0.393 | 0.416 | **0.417** | 0.387 | 229 | 434 | 248 | 268 | 248 |
| reddit | TGN v2 | 0.280 | 0.336 | **0.453** | 0.451 | 0.355 | 678 | 1392 | 938 | 1325 | 1317 |
| | DyRep v2 | 0.282 | 0.336 | **0.463** | 0.449 | 0.348 | 716 | 1402 | 1018 | 1116 | 937 |
| token | TGN v2 | 0.126 | 0.129 | 0.158 | **0.164** | 0.126 | 962 | 1819 | 1350 | 1848 | 1843 |
| | DyRep v2 | 0.126 | 0.132 | 0.150 | **0.159** | 0.133 | 974 | 1842 | 1333 | 1495 | 1386 |

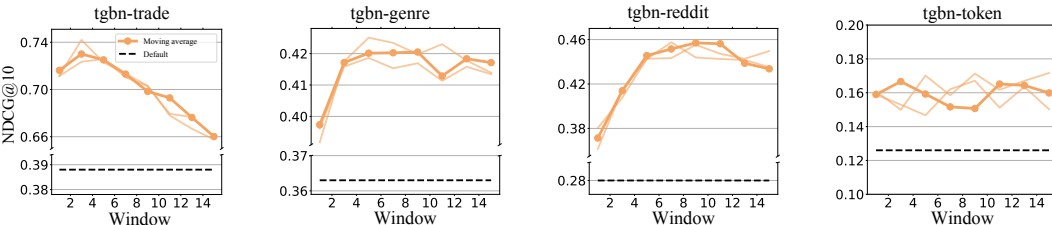

Figure 4: NDCG@10 versus Moving Average window size after one epoch of training using the TGNv2 model. Dashed lines indicate default TGNv2 performance without pseudo-labels.

figurations, where the vanilla model is trained for $X$ epochs. These configurations are selected such that the total training time of Default-X roughly matches or slightly exceeds that of the aggregation-based methods. This adjustment ensures a fairer comparison, since aggregation strategies typically require additional computations for model updates and label computations, which may slightly increase per-step overhead.

Even under this more favorable setup for the baseline, aggregation strategies consistently outperform the Default-X variants. These results confirm that the benefits of pseudo-label aggregation are not limited to faster convergence in terms of steps, but also translate directly into real-world time savings. Aggregation-based strategies enable more effective learning even under tight time budgets, making them especially suitable for scenarios with limited computational resources or where models must be frequently retrained in streaming environments.

Figure 3 shows NDCG@10 progression on tgbn-token, the most challenging dataset due to extreme sparsity. All aggregation strategies dramatically outperform Default, achieving orders of magnitude faster convergence. MA shows steepest improvement (short-term trends), HA demonstrates strong early performance (long-term patterns), while PF lags due to rigid assumptions in dynamic cryptocurrency settings.

## 4.3 ADDITIONAL STUDIES

**Dependence on Moving Average Window Size** The Moving Average (MA) aggregation strategy balances temporal responsiveness and stability through its window size $w$. Small windows prioritize recent interactions, enabling rapid adaptation but amplifying noise, while large windows emphasize historical trends, stabilizing predictions but potentially delaying response to shifts.

Figure 4 shows NDCG@10 versus window size across datasets after one epoch of training. For all considered window sizes and dataset, MA outperforms the Default approach. Performance initially

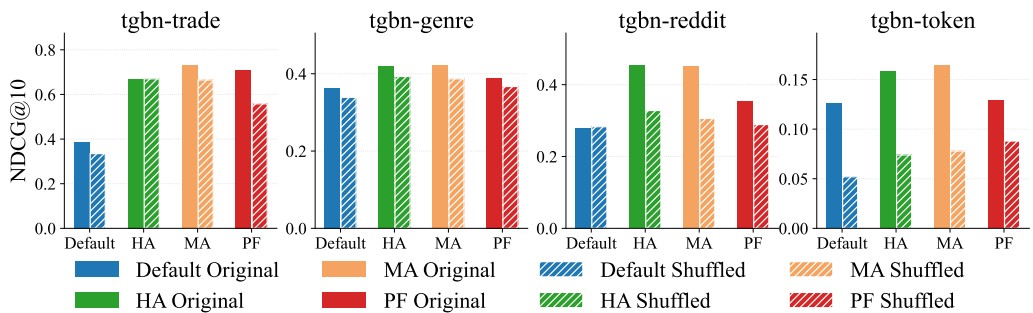

Figure 5: Impact of target shuffling across pseudo-label strategies after one epoch of training using the TGNv2 model.

improves with window size as pseudo-labels integrate sufficient context, then plateaus or declines as over-smoothing occurs. Optimal windows vary by dataset: trade networks (`tgbn-trade`) favor short windows ($w = 3$) for volatile patterns, while music genres (`tgbn-genre`) perform best with $w = 7$, reflecting gradual preference evolution.

**Order importance check with target shuffling.** Randomizing target chronology severely impacts all strategies, as evident from Figure 5, with absolute NDCG@10 reductions of 15–45% across datasets. This degradation demonstrates that temporal alignment between labels and node states is critical for effective learning. While performance drops vary by dataset (e.g., 31% for `tgbn-token` vs. 8% for `tgbn-trade`), all aggregations exhibit sensitivity to shuffled supervision. These results validate HA's core premise: leveraging temporally coherent historical signals is essential for mitigating sparse supervision in dynamic graphs.

**Detailed results.** Further experiments, including robustness studies, are reported in Appendix D, further support our findings. We demonstrate that the proposed aggregation methods maintain stability under varying levels of label noise, degrading gracefully as noise increases, while a vanilla loss provides sharper decline. Detailed tables with results equipped with standard deviations show both rapid quality improvements after a single epoch. A separate table details the test performance for an epoch selected via validation for training until convergence. There MA also provides competitive results.

## 5 CONCLUSION

We addressed a fundamental bottleneck in training Temporal Graph Networks (TGNs): the inefficiency caused by sparse supervision in real-world temporal data. Motivated by the observation that node-level preferences evolve gradually over time, we propose a simple yet effective pseudo-labeling strategy rooted in temporal aggregation of past labels. Our approach—comprising persistent forecasting, historical averaging, and moving averages—requires no additional parameters, incurs negligible computational overhead, and is agnostic to the underlying model architecture.

By employing pseudo-labeled batches, we convert otherwise idle training steps into meaningful updates, significantly accelerating convergence. Our theoretical analysis supports these findings, demonstrating that leveraging multiple plausible labels through historical aggregation improves the convergence speed.

Empirical results on the Temporal Graph Benchmark confirm the practical benefits of our method: up to $13\times$ faster convergence with no degradation in predictive performance when applied to the state-of-the-art TGNv2 model. These findings highlight the untapped potential of temporal consistency as a supervisory signal, offering a new direction for efficient learning on dynamic graphs.

Future work may explore adaptive aggregation strategies that learn optimal temporal weighting schemes or extend our method to other tasks such as link prediction or temporal graph completion. Ultimately, our approach contributes a lightweight yet powerful tool for overcoming supervision sparsity in temporal graph learning.

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

## A    RELATED WORK

**Temporal graph node classification**    Early approaches like TGAT Xu et al. (2020) used temporal attention for neighbor aggregation, while JODIE Kumar et al. (2019) learned coupled user-item embeddings via recurrent updates. The introduction of Temporal Graph Networks (TGNs) Rossi et al. (2020) with memory modules and continuous-time message passing marked a significant advance, enabling state-of-the-art performance on dynamic tasks. TGNv2 Tjandra et al. (2024) further improved expressivity by encoding node identities, addressing limitations in capturing persistent patterns. Recent work explores transformer architectures Yu et al. (2023) and lightweight MLP-based models Cong et al. (2023), balancing accuracy and efficiency.

**Sparsity of labels**    Scarcity of available labels appears even in regular graphs with no temporal component Zhan & Niu (2021). However, introducing pseudo-labels provides limited benefits for most existing methods, as methods suffer from information redundancy and noise in pseudo-labels Li et al. (2023). Moreover, these methods add a new training stage that employs pseudo-labels generated by a trained model, increasing overall training time. Label propagation through nodes is another approach Zhu (2005). While it allows for better introduction of the connections between nodes, more advanced methods are still required to unite them with graph convolutional networks Wang & Leskovec (2020).

**Generation of pseudo-labels**    Pseudo-labeling techniques have been widely adopted to address label scarcity. Semi-supervised approaches Lee et al. (2013); Xie et al. (2020) generate labels via self-training but suffer from confirmation bias in low-supervision regimes Arazo et al. (2020). Temporal knowledge graph (TKG) methods Han et al. (2023) interpolate entity distributions or missing facts but focus on triple completion, not node-level affinity prediction. Our key innovation lies in exploiting intrinsic temporal consistency through lightweight aggregation (e.g., moving averages), inspired by time series forecasting Hyndman & Athanasopoulos (2018). Unlike model-dependent pseudo-labeling Li et al. (2023), our method requires no auxiliary parameters, using historical interaction patterns to guide training during label-scarce periods. This aligns with streaming learning principles where historical baselines mitigate concept drift Ma et al. (2021).

Few works also considered the dynamics of user preferences Klenitskiy et al. (2024) and more general node classification problem Li et al. (2024) change over time. In both works, the authors come to the conclusion that for different datasets the dynamics ca be different: with some problems demonstrating close-to-constant user preferences, justifying label propagation and smoothing under true labels sparsity.

**Convergence for SGD.**    The convergence speed of Stochastic Gradient Descent (SGD) is fundamentally constrained by the variance of stochastic gradients, especially in the presence of label sparsity or noise. Classical results Shalev-Shwartz & Ben-David (2014); Bubeck (2015) show that, under natural assumptions, SGD converges at a rate of $\mathcal{O}(\sigma^2/T)$, where $T$ is the number of iterations, and $\sigma^2$ is the gradient variance. While approaches like SVRG Johnson & Zhang (2013) and Adam Kingma & Ba (2015) target variance reduction through optimization techniques, relatively less attention has been paid to variance reduction at the label level. While empirically, label smoothing and pseudo-labeling methods address supervision sparsity by generating soft targets based on model predictions or historical trends, they consider different problem statements and ignore the sequential nature of the data, also avoiding a theoretical analysis of the convergence speed and focusing on quality improvement, significantly increasing training time.

**Research gap**    Current temporal graph methods address sparsity via architectural changes Tjandra et al. (2024) or sampling Cong et al. (2023), neglecting pseudo-label-driven acceleration. While CAW-N Wang et al. (2021) uses temporal walks for induction, it introduces sampling overhead. GraphMixer Cong et al. (2023) improves efficiency but remains supervision-bound. Our work bridges this gap by demonstrating that simple temporal aggregates—requiring no new parameters—transform idle batches into productive training steps, leveraging temporal consistency for faster convergence without quality loss from both empirical and theoretical perspectives.

## B   PROOFS OF THE THEORETICAL RESULTS

For convenience, we repeat below the statements we aim to prove.

**Lemma 4.** *The expectation of $t_h$ is $\frac{u}{k}$ and the variance of $t_h$ is $u\frac{k-1}{k^2h} + u(1-u)\frac{1}{k^2}$.*

*Proof.*  We now prove Lemma 2.

By definition, $t_h$:
$$t_h = \eta_k \xi,$$
where $\xi$ is a Bernoulli random variable $\mathrm{Be}(u)$ and $\eta_k$ is a component of a multinomial random vector with equal probabilities $\frac{1}{k}$ and the total number of observations $h$, divided by $h$, as we aim to match the event type probability. Thus, it has the binomial distribution with parameters $\mathrm{Binomial}\left(h, \frac{1}{k}\right)$, divided by $h$. These two random variables are independent.

We'll derive the mean and the variance for the random variable that is the product of a Bernoulli and a Binomial random variable. Then, we'll scale the results by the coefficient $n$.

Let $B \sim \mathrm{Bernoulli}(u)$, so $B \in \{0, 1\}$ and $\mathbb{E}[B] = u$, $\mathrm{Var}(B) = u(1-u)$. Let $X \sim \mathrm{Binomial}(h, q)$, so $\mathbb{E}[X] = hq$, $\mathrm{Var}(X) = hq(1-q)$; $B$ and $X$ are independent.

Define the product random variable:
$$Y = BX.$$

Now let us derive the mean and the variance for $Y$.

We start with the mean. Because $B$ and $X$ are independent,
$$\mathbb{E}[Y] = \mathbb{E}[B]\mathbb{E}[X] = uhq.$$

The second moment for $Y$ is also easy to derive. Since $B^2 = B$, as it takes only 0 and 1 values,
$$Y^2 = B^2 X^2 = BX^2.$$

Again, using independence,
$$\mathbb{E}[Y^2] = \mathbb{E}[B]\mathbb{E}[X^2] = u\mathbb{E}[X^2].$$

For a binomial variable,
$$\mathbb{E}[X^2] = \mathrm{Var}(X) + (\mathbb{E}[X])^2 = hq(1-q) + h^2q^2.$$

Hence
$$\mathbb{E}[Y^2] = u\big[hq(1-q) + h^2q^2\big].$$

Now we are ready to obtain the variance of $Y$.
$$\mathrm{Var}(Y) = \mathbb{E}[Y^2] - \big(\mathbb{E}[Y]\big)^2 = u\big[hq(1-q) + h^2q^2\big] - u^2h^2q^2.$$

Simplifying this expression, we get:
$$\mathrm{Var}(Y) = uhq(1-q) + u(1-u)h^2q^2.$$

Going back to our original notation, we get the desired mean and variance:
$$\mathbb{E}[t_h] = \frac{1}{h}uh\frac{1}{k} = \frac{u}{k}.$$
$$\mathrm{Var}(t_h) = \frac{1}{h^2}\left(uh\frac{1}{k}\frac{k-1}{k} + u(1-u)h^2\frac{1}{k^2}\right) =$$
$$= u\frac{k-1}{k^2h} + u(1-u)\frac{1}{k^2}.$$

$\square$

**Theorem 5.** *We consider SGD in settings from Theorem 1. The following inequalities for the regret hold for a positive constant $c$ for the constants defined above:*

- *Under the assumptions of OHL, for the regret $R_T$ it holds:*

$$R_T \le \left(1 - \frac{u}{k}\right) \frac{u}{k} \frac{c}{\mu B} \frac{1 + \log T}{T} \le \frac{u}{k} \frac{c}{\mu B} \frac{1 + \log T}{T}.$$

- *Under the assumptions of HAL, for the regret $R_T$ it holds:*

$$R_T \le \left(\frac{k-1}{kh} + \frac{1-u}{k}\right) \frac{u}{k} \frac{c}{\mu B} \frac{1 + \log T}{T} \le$$

$$\le \frac{2}{\min(h, k)} \frac{u}{k} \frac{c}{\mu B} \frac{1 + \log T}{T}.$$

*Proof.* We now prove Theorem 3.

Settings of Theorem 1 hold, so the regret is bounded by:

$$R_T \le \frac{17\sigma^2}{\mu BT} \left(1 + \log T\right).$$

The variance $\sigma^2$ following the discussion in the main part of the paper has the form: $\sigma^2 = c \operatorname{Var}(t_h)$. Taking $\operatorname{Var}(t_h)$ from Lemma 2, we get $\operatorname{Var}(\sigma^2) = c \left(u \frac{k-1}{k^2 h} + u(1-u) \frac{1}{k^2}\right)$.

Plugging the expression for $\sigma^2$ into a general equation for regret for an arbitrary $h$ and $h = 1$, we get the first pair of the desired bounds in the Theorem.

Now, by construction

$$1 - \frac{u}{k} \le 1,$$

$$\frac{k-1}{kh} + \frac{1-u}{k} < \frac{1}{h} + \frac{1}{k} \le \frac{2}{\min(h, k)}.$$

There is a pair of rightmost upper bounds. □

## C ADDITIONAL EXPERIMENTS SETUPS

**Dataset properties** The dataset properties are listed in Table 2

Table 2: Statistics for used datasets. Density is the number of batches with non-zero labels divided by the total number of batches

| Dataset | Number of | | Density |
|---|---|---|---|
| | Nodes | Edges | |
| tgbn-trade | 255 | 337,224 | 1.30% |
| tgbn-genre 5% | 1,505 | 625,044 | 1.31% |
| tgbn-reddit 5% | 11,766 | 951,095 | 0.44% |
| tgbn-token 5% | 61,756 | 2,552,796 | 0.06% |

**Metrics.** We evaluate model performance using Normalized Discounted Cumulative Gain at rank 10 (NDCG@10) Järvelin & Kekäläinen (2002), which measures the ranking quality of the top-10 predicted affinities against ground-truth distributions. For example, in music genre prediction, NDCG@10 quantifies how well the model prioritizes genres a user is likely to engage with, based on their historical listening frequencies. Higher NDCG@10 indicates better model quality.

# D  ADDITIONAL EXPERIMENTS

**Noise factor**  An important consideration is the robustness of HAL to noisy historical labels, as real-world temporal graphs often contain imperfect supervision signals. Our empirical analysis demonstrates that the method maintains stability across varying noise levels, with performance degrading gracefully as noise increases rather than suffering catastrophic failure. The aggregation mechanism inherently provides some noise resilience by smoothing individual noisy observations across temporal windows, making the approach practical for real-world deployment where perfect labels are rarely available.

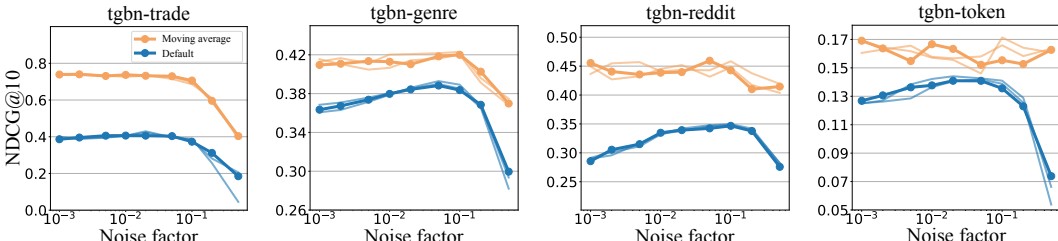

Figure 6: NDCG@10 versus noise factor after one epoch of training using the TGNv2 model.

**Target shuffle**  Table 3 demonstrates the critical importance of temporal alignment in pseudo-labeling strategies through a comprehensive target shuffling ablation study. When target chronology is randomized, all aggregation methods experience substantial performance degradation across datasets, with absolute NDCG@10 reductions ranging from 15% to 45%. The impact varies significantly by domain: cryptocurrency transactions (tgbn-token) show extreme sensitivity, while trade networks demonstrate greater resilience. This validates the fundamental premise that temporal consistency in node preferences is essential for effective pseudo-labeling in dynamic graphs.

Table 3: Impact of target shuffling across pseudo-label strategies.

| Dataset | Strategy | Test | | Valid | |
| --- | --- | --- | --- | --- | --- |
| | | Original | Shuffled | Original | Shuffled |
| tgbn-trade | Default | $0.388 \pm 0.006$ | $0.335 \pm 0.040$ | $0.405 \pm 0.008$ | $0.356 \pm 0.034$ |
| | HA | $0.669 \pm 0.005$ | $0.670 \pm 0.007$ | $0.737 \pm 0.002$ | $0.739 \pm 0.008$ |
| | MA | $0.729 \pm 0.010$ | $0.665 \pm 0.021$ | $0.813 \pm 0.015$ | $0.732 \pm 0.025$ |
| | PF | $0.710 \pm 0.003$ | $0.558 \pm 0.060$ | $0.800 \pm 0.001$ | $0.605 \pm 0.065$ |
| tgbn-genre 5% | Default | $0.363 \pm 0.005$ | $0.339 \pm 0.005$ | $0.369 \pm 0.005$ | $0.347 \pm 0.004$ |
| | HA | $0.419 \pm 0.003$ | $0.393 \pm 0.004$ | $0.422 \pm 0.002$ | $0.395 \pm 0.002$ |
| | MA | $0.421 \pm 0.003$ | $0.387 \pm 0.005$ | $0.421 \pm 0.001$ | $0.391 \pm 0.001$ |
| | PF | $0.389 \pm 0.003$ | $0.367 \pm 0.008$ | $0.387 \pm 0.002$ | $0.366 \pm 0.008$ |
| tgbn-reddit 5% | Default | $0.280 \pm 0.006$ | $0.283 \pm 0.013$ | $0.298 \pm 0.007$ | $0.308 \pm 0.019$ |
| | HA | $0.453 \pm 0.014$ | $0.328 \pm 0.018$ | $0.510 \pm 0.006$ | $0.408 \pm 0.018$ |
| | MA | $0.451 \pm 0.011$ | $0.306 \pm 0.014$ | $0.497 \pm 0.006$ | $0.390 \pm 0.014$ |
| | PF | $0.355 \pm 0.019$ | $0.289 \pm 0.002$ | $0.421 \pm 0.016$ | $0.358 \pm 0.004$ |
| tgbn-token 5% | Default | $0.126 \pm 0.004$ | $0.052 \pm 0.043$ | $0.123 \pm 0.009$ | $0.047 \pm 0.041$ |
| | HA | $0.158 \pm 0.029$ | $0.074 \pm 0.063$ | $0.253 \pm 0.022$ | $0.102 \pm 0.084$ |
| | MA | $0.164 \pm 0.024$ | $0.078 \pm 0.064$ | $0.266 \pm 0.009$ | $0.099 \pm 0.082$ |
| | PF | $0.129 \pm 0.016$ | $0.088 \pm 0.070$ | $0.217 \pm 0.017$ | $0.106 \pm 0.086$ |

# E  ADDITIONAL RESULTS

Table 4 demonstrates comprehensive performance comparisons across four TGB datasets, showing both NDCG@10 scores and training times after a single epoch.

Table 4: Quality and efficiency metrics after 1 training epoch. For more consistent comparison, we also include Default-X options, which correspond to the training of the vanilla TGNv2 for X epochs. Results averaged across 3 seeds.

| Dataset | Model | Strategy | NDCG@10 ↑ | | Time(s) ↓ |
|---|---|---|---|---|---|
| | | | Test | Val | |
| tgbn-trade | TGN v2 | Default | $0.388 \pm 0.006$ | $0.405 \pm 0.008$ | $23 \pm 2$ |
| | | Default-4 | $0.469 \pm 0.003$ | $0.499 \pm 0.003$ | $84 \pm 4$ |
| | | HA | $0.669 \pm 0.005$ | $0.737 \pm 0.002$ | $74 \pm 11$ |
| | | MA | $\mathbf{0.729} \pm 0.010$ | $\mathbf{0.813} \pm 0.015$ | $78 \pm 15$ |
| | | PF | $\underline{0.710} \pm 0.003$ | $\underline{0.800} \pm 0.001$ | $68 \pm 6$ |
| | Dyrep v2 | Default | $0.387 \pm 0.014$ | $0.403 \pm 0.016$ | $26 \pm 1$ |
| | | Default-4 | $0.448 \pm 0.001$ | $0.479 \pm 0.002$ | $102 \pm 2$ |
| | | HA | $0.668 \pm 0.001$ | $0.738 \pm 0.001$ | $91 \pm 7$ |
| | | MA | $\mathbf{0.734} \pm 0.003$ | $\mathbf{0.821} \pm 0.001$ | $94 \pm 18$ |
| | | PF | $\underline{0.711} \pm 0.003$ | $\underline{0.800} \pm 0.002$ | $81 \pm 16$ |
| tgbn-genre 5% | TGN v2 | Default | $0.363 \pm 0.005$ | $0.369 \pm 0.005$ | $177 \pm 6$ |
| | | Default-2 | $0.392 \pm 0.002$ | $0.399 \pm 0.003$ | $355 \pm 7$ |
| | | HA | $\underline{0.419} \pm 0.003$ | $\mathbf{0.422} \pm 0.002$ | $233 \pm 5$ |
| | | MA | $\mathbf{0.421} \pm 0.003$ | $\underline{0.421} \pm 0.001$ | $333 \pm 10$ |
| | | PF | $0.389 \pm 0.003$ | $0.387 \pm 0.002$ | $259 \pm 31$ |
| | Dyrep v2 | Default | $0.363 \pm 0.005$ | $0.368 \pm 0.006$ | $229 \pm 14$ |
| | | Default-2 | $0.393 \pm 0.002$ | $0.398 \pm 0.001$ | $434 \pm 12$ |
| | | HA | $\underline{0.416} \pm 0.002$ | $\underline{0.420} \pm 0.004$ | $248 \pm 20$ |
| | | MA | $\mathbf{0.417} \pm 0.001$ | $\mathbf{0.421} \pm 0.001$ | $268 \pm 37$ |
| | | PF | $0.387 \pm 0.003$ | $0.384 \pm 0.002$ | $248 \pm 22$ |
| tgbn-reddit 5% | TGN v2 | Default | $0.280 \pm 0.006$ | $0.298 \pm 0.007$ | $678 \pm 4$ |
| | | Default-2 | $0.336 \pm 0.003$ | $0.366 \pm 0.002$ | $1393 \pm 13$ |
| | | HA | $\mathbf{0.453} \pm 0.014$ | $\mathbf{0.510} \pm 0.006$ | $938 \pm 29$ |
| | | MA | $\underline{0.451} \pm 0.011$ | $\underline{0.497} \pm 0.006$ | $1325 \pm 7$ |
| | | PF | $0.355 \pm 0.019$ | $0.421 \pm 0.016$ | $1317 \pm 10$ |
| | Dyrep v2 | Default | $0.282 \pm 0.006$ | $0.299 \pm 0.006$ | $716 \pm 42$ |
| | | Default-2 | $0.336 \pm 0.001$ | $0.367 \pm 0.001$ | $1402 \pm 54$ |
| | | HA | $\mathbf{0.463} \pm 0.001$ | $\mathbf{0.513} \pm 0.002$ | $1018 \pm 127$ |
| | | MA | $\underline{0.449} \pm 0.011$ | $\underline{0.496} \pm 0.003$ | $1116 \pm 144$ |
| | | PF | $0.348 \pm 0.011$ | $0.415 \pm 0.008$ | $937 \pm 58$ |
| tgbn-token 5% | TGN v2 | Default | $0.126 \pm 0.004$ | $0.123 \pm 0.009$ | $962 \pm 20$ |
| | | Default-2 | $0.129 \pm 0.003$ | $0.133 \pm 0.004$ | $1819 \pm 33$ |
| | | HA | $\underline{0.158} \pm 0.029$ | $\underline{0.253} \pm 0.022$ | $1350 \pm 56$ |
| | | MA | $\mathbf{0.164} \pm 0.024$ | $\mathbf{0.266} \pm 0.009$ | $1848 \pm 4$ |
| | | PF | $0.126 \pm 0.016$ | $0.217 \pm 0.017$ | $1843 \pm 18$ |
| | Dyrep v2 | Default | $0.126 \pm 0.004$ | $0.124 \pm 0.008$ | $974 \pm 21$ |
| | | Default-2 | $0.132 \pm 0.003$ | $0.133 \pm 0.004$ | $1842 \pm 28$ |
| | | HA | $\underline{0.150} \pm 0.012$ | $\underline{0.255} \pm 0.005$ | $1333 \pm 31$ |
| | | MA | $\mathbf{0.159} \pm 0.012$ | $\mathbf{0.259} \pm 0.012$ | $1495 \pm 166$ |
| | | PF | $0.133 \pm 0.019$ | $0.221 \pm 0.016$ | $1386 \pm 158$ |

Table 5 shows the performance and efficiency of the considered approaches with and without aggregation. The NDCG@10 for all methods is very similar, with aggregation approaches slightly below the metric for the Default approach without aggregation. Time measurement spans from the start of training until convergence at the best validation epoch, inclusive of periodic validation/testing evaluations during training. This reflects real-world training scenarios where early stopping depends on validation performance, and runtime costs incorporate both learning and evaluation phases. Regard-

ing efficiency, both our metrics, Number of steps and Time, show improvements for all aggregation methods. The best aggregation method varies among datasets, but MA shows a more consistent improvement in efficiency and higher average quality.

Table 5: Quality and efficiency metrics until convergence. Time measurement spans from training start until the best validation epoch, including evaluation phases. Results averaged across 3 seeds.

| Dataset | Model | Strategy | NDCG@10 ↑ | | Time(s) ↓ |
| --- | --- | --- | --- | --- | --- |
| | | | Test | Val | |
| tgbn-trade | TGN v2 | Default | $0.735 \pm 0.012$ | $0.807 \pm 0.008$ | $5811 \pm 124$ |
| | | HA | $0.674 \pm 0.009$ | $0.745 \pm 0.011$ | $3249 \pm 89$ |
| | | MA | $\mathbf{0.741} \pm 0.006$ | $\mathbf{0.828} \pm 0.004$ | $\underline{887} \pm 25$ |
| | | PF | $0.715 \pm 0.008$ | $0.800 \pm 0.007$ | $3717 \pm 156$ |
| | DyRep v2 | Default | $0.728 \pm 0.015$ | $0.801 \pm 0.012$ | $5942 \pm 138$ |
| | | HA | $0.681 \pm 0.007$ | $0.751 \pm 0.009$ | $3187 \pm 95$ |
| | | MA | $\mathbf{0.745} \pm 0.008$ | $\mathbf{0.832} \pm 0.006$ | $\underline{923} \pm 31$ |
| | | PF | $0.719 \pm 0.011$ | $0.796 \pm 0.008$ | $3845 \pm 172$ |
| tgbn-genre 5% | TGN v2 | Default | $0.432 \pm 0.007$ | $0.437 \pm 0.006$ | $1071 \pm 22$ |
| | | HA | $\underline{0.422} \pm 0.005$ | $\underline{0.424} \pm 0.004$ | $\underline{223} \pm 8$ |
| | | MA | $\mathbf{0.420} \pm 0.004$ | $\mathbf{0.423} \pm 0.003$ | $264 \pm 12$ |
| | | PF | $0.388 \pm 0.006$ | $0.387 \pm 0.005$ | $238 \pm 15$ |
| | DyRep v2 | Default | $0.427 \pm 0.009$ | $0.432 \pm 0.008$ | $1158 \pm 28$ |
| | | HA | $\underline{0.418} \pm 0.006$ | $\underline{0.421} \pm 0.005$ | $\underline{241} \pm 11$ |
| | | MA | $\mathbf{0.416} \pm 0.005$ | $\mathbf{0.419} \pm 0.004$ | $278 \pm 14$ |
| | | PF | $0.384 \pm 0.007$ | $0.383 \pm 0.006$ | $252 \pm 18$ |
| tgbn-reddit 5% | TGN v2 | Default | $\underline{0.461} \pm 0.011$ | $0.496 \pm 0.009$ | $29890 \pm 485$ |
| | | HA | $\mathbf{0.461} \pm 0.008$ | $\mathbf{0.514} \pm 0.006$ | $\underline{1922} \pm 67$ |
| | | MA | $\mathbf{0.462} \pm 0.009$ | $0.513 \pm 0.007$ | $2547 \pm 88$ |
| | | PF | $0.384 \pm 0.012$ | $0.446 \pm 0.011$ | $17710 \pm 324$ |
| | DyRep v2 | Default | $\underline{0.456} \pm 0.013$ | $0.491 \pm 0.011$ | $30567 \pm 512$ |
| | | HA | $\mathbf{0.468} \pm 0.007$ | $\mathbf{0.518} \pm 0.005$ | $\underline{1845} \pm 72$ |
| | | MA | $\mathbf{0.467} \pm 0.010$ | $0.517 \pm 0.008$ | $2398 \pm 91$ |
| | | PF | $0.379 \pm 0.014$ | $0.441 \pm 0.013$ | $18256 \pm 348$ |
| tgbn-token 5% | TGN v2 | Default | $0.297 \pm 0.013$ | $0.312 \pm 0.015$ | $45580 \pm 892$ |
| | | HA | $0.295 \pm 0.011$ | $\underline{0.336} \pm 0.008$ | $\underline{6768} \pm 145$ |
| | | MA | $\mathbf{0.301} \pm 0.009$ | $\mathbf{0.337} \pm 0.007$ | $8910 \pm 198$ |
| | | PF | $0.269 \pm 0.014$ | $0.307 \pm 0.012$ | $14071 \pm 287$ |
| | DyRep v2 | Default | $0.289 \pm 0.016$ | $0.305 \pm 0.017$ | $47234 \pm 923$ |
| | | HA | $0.302 \pm 0.012$ | $\underline{0.341} \pm 0.009$ | $\underline{6521} \pm 152$ |
| | | MA | $\mathbf{0.308} \pm 0.010$ | $\mathbf{0.343} \pm 0.008$ | $8654 \pm 206$ |
| | | PF | $0.276 \pm 0.015$ | $0.312 \pm 0.013$ | $13789 \pm 295$ |

