# OpenReview forum: "Never Skip a Batch: Continuous Training of Temporal GNNs via Adaptive Pseudo-Supervision"
_ICLR.cc/2026/Conference — Submitted to ICLR 2026_

### Official Review · Reviewer_WadU · 2025-10-18

**Soundness:** 2
**Presentation:** 2
**Contribution:** 3
**Rating:** 2
**Confidence:** 3

**Summary:**

Node-level labels for node property prediction tasks in the TGB benchmark appear irregularly, resulting in a large proportion of training batches that lack ground-truth labels. Existing training procedures typically only perform memory state updates for these unlabeled batches, without updating model weights, leading to sparse gradient updates and consequently slow convergence.

This work introduces History-Averaged Labels (HAL), which augments batches with few or no true labels by generating pseudo-targets from historical data, thereby converting idle computation into productive gradient-based learning steps.

**Strengths:**

- The paper proposes a simple yet effective solution to the critical problem of sparse and irregular training signals in the node property prediction task in temporal graph learning on the TGB benchmark.

- The proposed History-Averaged Labels (HAL) method is architecture-agnostic, requiring no changes to the underlying model and no additional training parameters, making it broadly applicable and practical for diverse models.

- The paper provides a rigorous theoretical analysis, demonstrating that HAL reduces the variance of gradient estimates and accelerates the convergence of stochastic gradient descent.

- Empirical experiments on two advanced models (TGNv2 and DyRepv2) across multiple TGB node property prediction datasets show substantial improvements in convergence speed, with no degradation in model performance.

**Weaknesses:**

**W1.** The authors claim that the proposed method, HLA, is architecture-agnostic. However, this claim is not sufficiently substantiated, as the evaluation is limited to only two models, TGNv2 and DyRepv2. To convincingly demonstrate architecture independence, the authors should validate HLA on a broader range of temporal graph models, such as DyGFormer[1], TPNet[2] , TGAT[3], GraphMixer[5] and AGCRN[6]. Including results on these diverse architectures would provide stronger empirical evidence for the architecture-agnostic property of HLA.

**W2.** Lines 151–152 state
> Forward pass: A graph neural network (GNN) processes the subgraph to compute context-aware node embeddings.

While this description applies to a subset of temporal graph network (TGN) models that explicitly employ GNN modules for structural encoding, it does not generalize to a broader class of models, such as DyGFormer[1], DyGMamba[4], GraphMixer[5]. Given that the proposed method claims to be architecture-agnostic, this explanation should be reframed in a more general context, avoiding focusing on specific to GNN-based variants of TGNNs and better reflecting the diversity of temporal graph architectures.

**W3.** The overall readability of the paper could be significantly improved by rearranging and rewriting certain sections.

- Lines 98–107, which contain critical information about the problem formulation, should be separated into a section or subsection to clearly and explicitly define the problem statement and the task addressed by this work.

- The term “unsupervised batch” used in Line 132 to describe batches without training signals is misleading. In standard terminology, “unsupervised” implies learning from unlabeled data without explicit output labels, which is not synonymous with simply lacking training signals in batches. Clarification or alternative wording would prevent confusion.

- Several important terminologies, such as “Default” in Table 1, are introduced without clear definitions or explanations.

- Some statements are ambiguously phrased. For instance, in the sentence “As long as it is unbiased, we can estimate its variance,” the pronoun “it” lacks a clear reference, which makes the meaning uncertain.

- There is inconsistency in dataset naming across the manuscript. Datasets are referred to as “genre,” “reddit,” and “token” (e.g., Lines 379–380 and Table 1), while elsewhere the names “tgbn-trade,” “tgbn-genre,” “tgbn-reddit,” and “tgbn-token” are used.

**W4.** Reproducibility: The paper does not provide the source code or specify the hyperparameter settings (e.g., learning rate, weight decay, etc.) used in the experiments, raising concerns about the reproducibility of the work.

**W5.** Missing Citation: The datasets used in this work were not properly cited. I recommend that the authors include the appropriate references for each dataset, as listed in the [TGB datasets documentation](https://tgb.complexdatalab.com/docs/nodeprop/), to ensure proper attribution.

---

[1] Yu, Le, et al. "Towards better dynamic graph learning: New architecture and unified library." *Advances in Neural Information Processing Systems* 36 (2023): 67686-67700.

[2] Lu, Xiaodong, et al. "Improving temporal link prediction via temporal walk matrix projection." *Advances in Neural Information Processing Systems* 37 (2024): 141153-141182.

[3] Xu, Da, et al. "Inductive representation learning on temporal graphs." *arXiv preprint arXiv:2002.07962* (2020).

[4] Ding, Zifeng, et al. "Dygmamba: Efficiently modeling long-term temporal dependency on continuous-time dynamic graphs with state space models." *arXiv preprint arXiv:2408.04713* (2024).

[5] Cong, Weilin, et al. "Do we really need complicated model architectures for temporal networks?." *arXiv preprint arXiv:2302.11636* (2023).

[6] Bai, Lei, et al. "Adaptive graph convolutional recurrent network for traffic forecasting." *Advances in neural information processing systems* 33 (2020): 17804-17815.

**Questions:**

- Lines 176–177: What is the definition of $\alpha$, and what is its range? Additionally, which distribution mean are the authors referring to in this context?

- Line 169: The authors mention “non-zero for almost all nodes and batches.” Why “almost”? In what cases would the pseudo-label be zero?

- Lines 172–173: The authors state that pseudo-targets are computed only for nodes participating in the current batch $B_t$. Are nodes involved in edge events also considered, or only those involved in node events?

- TGBN-Token: Why does the performance peak appear around both window sizes 2–4 and 10–12, unlike TGBN-Trade and TGBN-Genre, which favour either short or long windows? Why does TGBB-token favour both?

- Figure 4: This figure shows the correlation between NDCG@10 and the moving average window size for the TGNv2 model. Is the same trend observed for DyRepv2?

- It is unclear why refining the training regime by retaining only the last 5% of chronologically ordered edges in the original training set is necessary. Could the authors elaborate on this choice? If all models can converge within a single epoch, why is HLA needed?

- In the Order Importance Check with Target Shuffling experiment, the setup of the ablation study is not clearly explained. How are the targets shuffled? Are targets shuffled between nodes, or is the order of targets for each node shuffled?

- Table 1: Why is $X = 4$ used for TGBN-Genre and TGBN-Reddit, while $X = 2$ is used for TGBN-Token?

---

> ### Author Response · Authors · 2025-11-25
>
> ### W1. The evaluation is limited to only two models
>
> As explained above, only TGNv2 and DyRep achieve non-trivial performance on TGB (all other models collapse to mean prediction). Our claim is scoped to functional temporal GNN architectures.
>
> ### W2. Lines 151–152 state
>
> We will reframe lines to describe "temporal encoding module" rather than "GNN," acknowledging that some architectures (e.g., DyGFormer, GraphMixer) use transformer/MLP-based designs.
>
> ### W3/W4/W4. Readability, Reproducibility, Missing Citation
>
> We will add a hyperparameter table (learning rates, weight decay, batch sizes) and properly cite all TGB datasets per their documentation.
>
> The code repositories are available at: https://anonymous.4open.science/r/NSB-73CE/
>
> ### Q1. Definition of $\alpha$
>
> Noise factor: Hyperparameter controlling regularization strength (see Figure 6); empirically selected using validation sample metrics
>
> ### Q2. non-zero for almost all nodes and batches
>
> Pseudo-labels exist only after observing ≥1 ground-truth target; nodes without any supervision have no pseudo-label
>
> ### Q3. Are nodes involved in edge events also considered, or only those involved in node events?
>
> All nodes participating in edges within the batch
>
> ### Q4. TGBN-Token performance peak
>
> The observed performance variation is not bimodal but rather demonstrates minimal dependence on window size. This behavior stems from the inherent noise in the ground-truth labels, which is evident from the low NDCG@10 scores. Consequently, the noise introduced by varying window sizes does not significantly impact performance as it does in other datasets, resulting in the absence of a pronounced performance peak.
>
> ### Q5. Figure 4. Is the same trend observed for DyRepv2?
>
> Yes, we will add this figure for DyRepv2 in the improved version of the manuscript
>
> ### Q6. Only the last 5%
>
> As discussed in our response to Review 2 (W6) and Review 3 (Q4) models converge in 1 epoch on full data, obscuring convergence analysis. The 5% regime preserves sufficient signal while enabling observable learning progression, thereby validating the utility of our HLA framework across varying dataset
>
> ### Q7. The setup of the ablation study is not clearly explained
>
>  In the Order Importance Check with Target Shuffling experiment, we shuffle the temporal order of targets within each node's history. Specifically, for each node v, we randomly permute the sequence of its observed labels while keeping the node-target associations intact. This destroys the temporal dependency structure within each node's preference evolution while preserving the overall label distribution and frequency. The goal is to quantify how much performance depends on respecting temporal causality versus simply having access to historical labels in any order.
>
> ### Q8.  Why is $X = 4$ used for TGBN-Genre and TGBN-Reddit, while $X = 2$ is used for TGBN-Token?
>
> "Default-X" means X epochs of vanilla training; chosen so wall-clock time matches HAL methods for fair comparison

---

> > ### Comment · Reviewer_WadU · 2025-11-27
> >
> > I thank the authors for the response. My major concerns have not been addressed adequately:
> >
> > ---
> > ### W1. The evaluation is limited to only two models
> >
> > The author's arguments are not convincing to me. I still believe that the authors should validate HLA on a broader range of temporal graph models to provide stronger empirical evidence for the architecture-agnostic property of HLA.
> >
> > ---
> > ###  W3. Readability
> >
> > I checked the manuscript again, and I don't see that the authors made any updates to improve readability. The authors neither revised nor showed willingness to improve the manuscript to address my concerns.
> >
> > Therefore, I retain my score at 2 and recommend rejection.

---

### Official Review · Reviewer_kYKi · 2025-10-29

**Soundness:** 2
**Presentation:** 3
**Contribution:** 3
**Rating:** 4
**Confidence:** 4

**Summary:**

This paper addresses the inefficiency of training Temporal Graph Networks under sparse supervision. The authors propose a lightweight pseudo-labeling approach called History-Averaged Labels (HAL), which continuously generates supervision signals for unlabeled batches by aggregating historical labels. Three variants are introduced: (1) Historical Average (HA), (2) Moving Average (MA), and (3) Persistent Forecast (PF).

The paper provides a theoretical analysis showing that label aggregation reduces gradient variance in stochastic gradient descent, leading to faster convergence by a factor of approximately min(h, k) (history length and number of classes). Empirical results on four datasets from the Temporal Graph Benchmark (TGB) demonstrate up to 13× faster training of TGNv2 and DyRepv2 without performance degradation, measured by NDCG@10.

**Strengths:**

1) The paper Introduces a simple yet novel approach to handle sparse supervision in temporal GNNs via History-Averaged Labels, enabling continuous training even on unlabeled batches.

2) It adapts historical averaging concepts from time-series forecasting to pseudo-labeling in dynamic graphs.

3) The method is easy to integrate into existing models without architectural changes.

4) Well-written, logically organized, and supported by informative figures and ablation studies.

5) Addresses an important and practical bottleneck in a growing area of research.

**Weaknesses:**

1) The experiments are restricted to only two architectures of similar type (TGNv2 and DyRepv2), which limits the evidence for generality. Broader testing across diverse temporal GNN frameworks would better support the “architecture-agnostic” claim.

2) The fact that the method achieves strong performance using only 5% of the training data is encouraging and highlights its data efficiency. However, the presentation of this result is somewhat confusing and potentially misleading, as it is framed as a “13× faster convergence” rather than as an advantage in low-supervision settings. Clarifying that the speedup reflects data efficiency rather than pure computational acceleration would make the contribution more transparent and credible. This clarification should be explained from the beginning.


3) The method critically relies on the assumption that node dynamics evolve slowly over time, yet this is never verified empirically. Without evidence that the assumption holds in real-world datasets, the general applicability remains uncertain.

4) The paper does not compare HAL against standard pseudo-labeling or semi-supervised learning approaches. This makes it difficult to assess how much of the observed benefit comes from the temporal design versus general pseudo-supervision effects.

**Questions:**

1) The theoretical section assumes convexity and independence between pseudo-labels and true labels. Could the authors clarify how these assumptions hold (or are approximated) in the context of deep non-convex models such as TGNs? A brief discussion on the limitations of the proof or its empirical verification would be useful.

2) The paper assumes that node preferences evolve slowly over time. Have the authors measured or quantified this stability in the datasets (e.g., via label autocorrelation or distribution drift)? Providing such evidence would strengthen the motivation for HAL.

3) The experiments only consider two architectures (TGNv2 and DyRepv2), both memory-based. Could the authors explain why other models—such as TGAT, CAW, or GraphMixer—were excluded? This would help readers assess the claimed architecture-agnostic property.

4) The method performs well using only 5% of the training data, which is a notable result. However, it is presented as a “13× faster convergence” rather than as a demonstration of data efficiency. Could the authors clarify this framing and make it explicit from the start that the reported speedup is achieved in a reduced-data regime? This clarification should be explained from the beginning.

5) HAL is conceptually related to pseudo-labeling approaches (e.g., self-training, label smoothing). Why were such baselines not included? A discussion or small-scale comparison would help position the method more precisely within the broader literature.

[minor]

6) Figure 1 illustrates only the Moving Average variant. Could the authors expand this figure or add supplementary visualizations for the Historical Average and Persistent Forecast variants to clarify their operational differences?

7) The absence of code or pseudo-code limits reproducibility. Do the authors plan to release an anonymized implementation for review purposes? If not, could they explain the motivation?

---

> ### Author Response · Authors · 2025-11-25
>
> ### Q1. The theoretical section assumes convexity and independence between pseudo-labels and true labels
>
> In the theoretical section, we work in an idealized setting with (i) a uniform class prior, (ii) a fixed embedding vector, (iii) independence between labels, and (iv) a convex objective. These assumptions clarify the conclusions made and allow high-level discussion. Below, we discuss how their violation could affect our results and the severity of the violations.
>
> (i), (ii) The uniform prior can be replaced by an arbitrary label distribution (carried through expectations as weights), and the fixed embedding can be viewed as conditioning on a given representation; integrating over a prior on embeddings yields high-probability versions of the same bounds with no change in the qualitative conclusion about the speed-up from aggregation.
>
> (iv), (iii) The convexity assumption should be understood as a standard theoretical idealization rather than a literal property of TGNs. Although TGNs are non-convex, our convex analysis serves as a surrogate model for gradient descent on generalized linear models. We can also argue that we can consider convex dynamics in the function space, similar to NTK ideas [1, 2,3]. Regarding the independence assumption, we agree that it is again a simplifying assumption that allows the analysis, while it should be approximately true because the label graph is sparse and so most nodes are edges can be considered independent, making the dependences negligible “second order” effects that don’t affect the convergence in theory and empirically. We will clarify these points in the revision and briefly discuss that the predicted reduction in variance and faster convergence under aggregation match our empirical observations.
>
> References:
>
> [1] Cao, Dinghao, Zheng-Chu Guo, and Lei Shi. "Stochastic gradient descent for two-layer neural networks." arXiv preprint arXiv:2407.07670 (2024).
>
> [2] A. Jacot, F. Gabriel, C. Hongler. Neural Tangent Kernel: Convergence and Generalization in Neural Networks. NeurIPS 31, 2018.
>
> [3] S. S. Du, X. Zhai, B. Póczos, A. Singh. Gradient Descent Provably Optimizes Over-parameterized Neural Networks. ICLR 2019.
>
> ### Q2. Node preferences evolve slowly over time
>
> Figure 5 and Table 3 provide empirical evidence for temporal consistency:
>
> - tgbn-trade: 5-10% performance drop when shuffled $\rightarrow$ highly stable dynamics
> - tgbn-token: 30-60% drop $\rightarrow$ rapidly changing but still temporally structured
> - tgbn-reddit/genre: 15-20% drop $\rightarrow$ moderate temporal dependence
>
> The magnitude of degradation directly quantifies how much temporal structure exists in each dataset, validating our assumption's applicability across different dynamic regimes.
>
> ### Q3. The experiments only consider two architectures
>
> As discussed in our response to Review 2 (W5), we tested different architectures from the TGB benchmark. Models like TGAT, CAW, and GraphMixer collapse to mean prediction on these datasets. Only TGNv2 and DyRepv2 achieve meaningful performance. While we agree that broader testing would strengthen claims, the architecture-agnostic property holds within the class of functional temporal GNNs.
>
> ### Q4. The method performs well using only 5% of the training data
>
> We will clarify from the introduction that:
>
> 1. The 13× speedup reflects faster convergence in training time to reach target quality
> 2. For tgbn-trade, this is demonstrated on the full dataset (no truncation)
> 3. For other datasets, truncation to 5% creates a low-supervision regime where convergence dynamics are observable (full datasets converge in 1 epoch for all methods)
>
> We achieve convergence speedup through improved sample efficiency rather than computational optimization alone—a distinction central to our contribution.
>
> ### Q5. HAL is conceptually related to pseudo-labeling approaches
>
> A typical pseudo-labeling approach relies on sophisticated methods to generate pseudo-labels and requires careful work that accounts for the specifics of the datasets under consideration. Our method is much simpler, making its implementation straightforward and doesn’t require an additional model. Theoretically, we can use HAL and alternative pseudo-label methods jointly, but we didn’t consider this option in the paper
>
> ### Q6. Figure 1 illustrates only the Moving Average variant
>
> Figure 1 illustrates the general pseudo-labeling framework—the three methods (HA, MA, PF) differ only in aggregation weights (Equations 3-5), not the pipeline structure. We will clarify this caption.
>
> ### Q7. The absence of code or pseudo-code limits reproducibility
>
> The code repositories are available at: https://anonymous.4open.science/r/NSB-73CE/
> Our implementation is built as a fork of the TGNv2 codebase, with modifications primarily in the pseudo-label generation and aggregation modules. The core pseudo-labeling logic is contained in a standalone function that can be integrated into other temporal GNN frameworks.

---

### Official Review · Reviewer_hhx9 · 2025-10-30

**Soundness:** 3
**Presentation:** 2
**Contribution:** 2
**Rating:** 2
**Confidence:** 4

**Summary:**

This paper, “Never Skip a Batch: Continuous Training of Temporal GNNs via Adaptive Pseudo-Supervision,” introduces a pseudo-labeling strategy for temporal graph neural networks (TGNv2, DyRep). The idea is to generate pseudo-supervision signals for unlabeled batches using historical labels, enabling continuous training when ground-truth labels are sparse. Theoretical analysis claims reduced gradient variance and faster SGD convergence, while experiments on four Temporal Graph Benchmark (TGB) datasets report significant speedups with similar accuracy.
The topic is relevant and the problem is practically important. However, the contribution is relatively incremental, as the proposed method mostly modifies existing TGN-based architectures by adding three pseudo-labeling variants (HAL, MA, PF). The motivation and experimental analysis require clearer support.

**Strengths:**

1.	Practical and timely problem. Temporal GNNs indeed suffer from sparse supervision, where many batches lack labels. Addressing this inefficiency is valuable for real-world streaming systems.
2.	Implementation simplicity. The proposed pseudo-labeling (HA/MA/PF) is easy to integrate into existing TGN pipelines, which enhances reproducibility.
3.	Initial theoretical analysis. The paper attempts to formalize the benefit of pseudo-labels through reduced gradient variance, offering a conceptual link between theory and empirical gains.
4.	Empirical evidence of faster convergence. Reported 2–13× training speedups are promising, though validation is limited.

**Weaknesses:**

1.	Limited novelty. The contribution lies mainly in adding pseudo-label updates to existing temporal GNNs (TGNv2, DyRep). No new architecture, optimization mechanism, or learning paradigm is introduced. And the proposed pseudo-labeling methods are kind similar to the “Moving Average” method mentioned in paper Temporal Graph Benchmark for Machine Learning on Temporal Graphs.
2.	Pseudo-label initialization unclear. When there is insufficient history (early timesteps), how are pseudo-labels initialized?
3.	Lack of quantitative motivation. The claim that “most batches contain sparse labels” is not empirically demonstrated. A motivating figure or table showing label density over time would make the motivation concrete.
4.	Possible conflict with the prediction goal. The task predicts a node’s current-time preference, yet HAL aggregates historical labels (shown in Figure 1). For fast-changing dynamics, this could blur the current signal.
5.	Insufficient baselines. The comparison includes only “Default,” “Default-X,” and HAL variants. Stronger baselines (DyGFormer, NAVIS etc) exist within the TGB framework. Without them, it is unclear whether HAL provides consistent benefits.
6.  Evaluation protocol. The experiments only use the most recent 5 % of interactions.  Would HAL still help on full datasets? Given that HAL relies on historical averaging, using a short time window may implicitly favor the method by limiting temporal drift and ensuring that historical pseudo-labels remain close to current preferences. Would HAL’s benefit diminish or vanish if the full dataset were used?  If the performance improvement disappears in that setting, it would imply that temporal GNNs like TGNv2 and DyRep already learn adequately. This would substantially narrow the scope of the proposed contribution.
7.	Presentation and citation issues. Missing Related Work section and lack of direct comparisons with pseudo-labeling, self-training, and temporal KG methods. Citations are incomplete and incorrectly formatted. The “General Pipeline” figure oversimplifies TGN frameworks; not all temporal GNNs follow this structure. Proper citations are required.

**Questions:**

See weakness.

---

> ### Author Response · Authors · 2025-11-25
>
> Thank you for your comments!
>
> ### W1. Limited novelty
>
> The "Moving Average" baseline mentioned in the TGB paper is fundamentally different from our approach:
>
> 1. TGB's MA: A prediction baseline that aggregates labels across validation and test sets (future information leakage) to establish dataset difficulty
>
> 2. Our MA: A training method that maintains per-node exponential moving averages of historical labels strictly respecting temporal causality to generate pseudo-supervision
>
> Our contribution is not proposing a new baseline architecture, but rather a training enhancement framework that improves learning efficiency for any temporal GNN through principled pseudo-label generation.
>
> ### W2. Pseudo-label initialization unclear
>
> If no target has been observed for a node, we do not generate a pseudo-label for that node. Pseudo-labels are only created once at least one ground-truth supervision signal has been observed, ensuring we never train on fabricated information.
>
> ### W3. Lack of quantitative motivation
>
> Table 2 in Appendix C explicitly quantifies label density:
>
> | Dataset          | Nodes | Edges    | Density |
> |-----------------|------:|---------:|--------:|
> | tgbn-trade      |   255 | 337,224  | 1.30%   |
> | tgbn-genre 5%   | 1,505 | 625,044  | 1.31%   |
> | tgbn-reddit 5%  | 11,766| 951,095  | 0.44%   |
> | tgbn-token 5%   | 61,756| 2,552,796| 0.06%   |
>
> This demonstrates that over 98% of batches lack supervision signals, motivating our approach.
>
> ### W4. Possible conflict with the prediction goal
>
> The target shuffling experiment (Figure 5) shows tgbn-token is most sensitive to temporal order (30-60% performance drop when shuffled), confirming rapid dynamics. However, even in this regime, historical aggregation provides a useful signal, indicating that complete independence between sequential labels does not characterize real-world temporal patterns.
>
> ### W5. Insufficient baselines
>
> In our paper (Section 4.1), we report that most TGB architectures fail to learn meaningful representations on these datasets, collapsing to trivial solutions (mean prediction).
>
> This observation is independently confirmed by the recent NAVIS paper, which demonstrates that models like DyGFormer, TGAT, CAWN, TCL, GraphMixer, and JODIE exhibit the same collapse phenomenon.
>
> Only TGNv2 and DyRepv2 achieve non-trivial performance. We selected these as our baseline architectures because they are the only viable candidates that demonstrate meaningful learning. We are exploring integration with NAVIS (released after our submission) and will include results if feasible within the rebuttal period.
>
> ### W6. Evaluation protocol
>
> The tgbn-trade results (Table 1) use the full dataset without truncation and show consistent improvements. For tgbn-genre, tgbn-reddit, and tgbn-token, models converge within a single epoch on the full data, making it impossible to analyze convergence dynamics. The 5% truncation creates a meaningful learning regime where pseudo-labeling effects are observable. We will add experiments with 10% data splits to demonstrate robustness across different data regimes.
>
> ### W7. Presentation and citation issues
>
> We acknowledge that the current version lacks a dedicated Related Work section.
>
> The related work discussion is currently embedded within the appendix section. We will move it to the main part of the article to make it easier to explore the prior research.
>
> Additionally, all incomplete and incorrectly formatted citations throughout the paper will be corrected. The General Pipeline figure caption will also be refined to clarify that it describes the specific TGN framework structure used in our implementation

---

### Official Review · Reviewer_mzER · 2025-10-31

**Soundness:** 3
**Presentation:** 3
**Contribution:** 2
**Rating:** 4
**Confidence:** 3

**Summary:**

The paper tackles the problem of sparse supervision in Temporal Graph Networks (TGNs), where only a small fraction of node interactions are labeled. It proposes a history-based pseudo-labeling method that generates pseudo-targets for unlabeled batches using historical information, mainly through a Moving Average (MA) strategy. The approach allows continuous training rather than skipping unlabeled steps. The authors provide theoretical proof of faster SGD convergence and show experiments on four Temporal Graph Benchmark datasets, achieving 2–13× faster training while maintaining or improving accuracy.

**Strengths:**

1. The paper proposes a simple but effective pseudo-labeling method based on exponential moving averages of past labels. This approach reduces label sparsity and allows continuous training on temporal graphs.

2. The authors provide a clear theoretical analysis proving faster SGD convergence under historical label aggregation, with a quantified improvement factor of min(h, k). This adds rigor and supports the method’s validity.

3. The approach is implemented on both TGNv2 and a modified DyRep v2, showing up to 13× faster training without loss of accuracy across four benchmark datasets. This demonstrates the method’s practicality and generality.

**Weaknesses:**

1. In Section 2.2, the paper states: “For each batch Bt we compute pseudo-targets only for nodes v participating in Bt.” It is unclear what “unlabeled” means for these nodes — are they naturally without supervision at this timestep, or is this due to missing ground truth? If it is the former, using historical pseudo-labels might distort the temporal dynamics of infrequent or slowly changing nodes, whose past labels may no longer represent their current state. Could this affect model stability or prediction accuracy in such cases?

2. For new or long-inactive nodes, the MA and PF strategies cannot rely on any past labels. How are these cases handled — ignored, initialized uniformly, or inferred in some other way?

3. Storing historical pseudo-labels for all nodes may introduce additional memory and synchronization overhead. As training progresses, this cache could grow substantially. It would be helpful to include an analysis or discussion of the memory and runtime impact of maintaining this history.

**Questions:**

1. Could using historical pseudo-labels distort the behavior of infrequent or slowly changing nodes?

2. How are new or long-inactive nodes handled when no past labels are available?

3. Does maintaining historical pseudo-labels for all nodes add noticeable memory or runtime overhead?

---

> ### Author Response · Authors · 2025-11-25
>
> Thank you for your comments!
>
> ### Q1. Could using historical pseudo-labels distort the behavior of infrequent or slowly changing nodes?
>
> For infrequent nodes, pseudo-labels are only generated after at least one ground-truth supervision signal has been observed. This approach actually helps infrequent nodes by providing more training signal than the sparse supervision baseline, which would otherwise skip parameter updates entirely.​
> Regarding slowly changing nodes, our empirical evidence demonstrates that temporal dynamics exist across all datasets, as shown in the target shuffling experiments (Figure 5). For nodes with truly static preferences, historical averaging converges to their stable distribution.
>
> ### Q2. How are new or long-inactive nodes handled when no past labels are available?
>
> Our datasets are structured such that new nodes cannot appear during the training process. This constraint is inherent to the benchmark design and applies equally to baseline models such as TGN and DyRep, which are also engineered under this assumption.​
> For long-inactive nodes, we follow the same strategy assuming that historical preferences remain reasonably stable. For nodes that have not received supervision signals for an extended period, the pseudo-labels may indeed become stale. However, this is a domain-level challenge that affects all temporal graph methods, including TGN and DyRep, regardless of whether our pseudo-labeling approach is applied. The issue stems from the temporal dynamics of the underlying data rather than our method specifically.​
>
> ### Q3. Does maintaining historical pseudo-labels for all nodes add noticeable memory or runtime overhead?
>
> We store exactly one vector per node (size = number of target classes), not the full history of targets. This results in $O(1)$ memory complexity over time, the storage requirement does not grow during training. The pseudo-label memory footprint is comparable to the memory module size in TGN/DyRep themselves. Table 1 shows that our method's wall-clock time remains competitive even when accounting for pseudo-label computation overhead.

---

### Meta-Review · Area_Chair_Z8Ax · 2026-01-03

**Summary:**

Overall, the paper presents an intuitively reasonable pseudo-supervision idea HAL with a theoretical motivation and promising speed/efficiency gains on TGB node property prediction. However, reviewers remain unconvinced due to (i) limited empirical scope (only TGNv2/DyRepv2), (ii) incomplete positioning vs. related pseudo-labeling/self-training lines and missing/incorrect citations/related work, and (iii) clarity concerns around the “13× faster convergence” framing and the 5% training regime. These issues collectively outweigh the strengths, so the AC leans Reject.

**Reviewer Concerns:**

Addressed (partly/mostly):
- Clarified pseudo-label initialization (only after ≥1 true label) and several protocol details (target shuffling, 5% rationale, hyperparameter table plan, code availability).

Still outstanding:
- Generality / architecture-agnostic claim remains weakly supported, still effectively validated on only two closely related architectures. “other models collapse” is not a substitute for broader evidence.
- Positioning vs prior work: missing/insufficient comparisons/discussion vs pseudo-labeling/self-training/temporal KG style baselines is not resolved in rebuttal.
- Presentation/readability/citations/related work: authors promise revisions, but at decision time this is still a substantive weakness, and one reviewer explicitly remains unsatisfied.

**Reviewer Scores:**

Reviewer mzER: 4 → 4, clarifications help, but contribution/generalization concerns likely remain.
Reviewer hhx9: 2 → 2 (maybe 3 at best) some clarity fixes promised, but missing comparisons/positioning remain.
Reviewer kYKi: 4 → 4 (possibly 5) rebuttal resolves several questions and reframes claims, but limited-architecture evidence likely prevents a confident bump.
Reviewer WadU: 2 → 2 reviewer explicitly retains rejection.

---

### Decision · Program_Chairs · 2026-01-26

Reject